# Factors Associated with the Prevalence and Severity of Menstrual-Related Symptoms: A Systematic Review and Meta-Analysis

**DOI:** 10.3390/ijerph20010569

**Published:** 2022-12-29

**Authors:** Risa Mitsuhashi, Akemi Sawai, Kosuke Kiyohara, Hitoshi Shiraki, Yoshio Nakata

**Affiliations:** 1Graduate School of Comprehensive Human Sciences, University of Tsukuba, 1-1-1 Tennodai, Tsukuba 305-8574, Japan; 2Research Institute of Physical Fitness, Japan Women’s College of Physical Education, 8-19-1 Kitakarasuyama, Setagaya-ku 157-8565, Japan; 3Department of Food Science, Faculty of Home Economics, Otsuma Women’s University, 12 Sanban-cho, Chiyoda-ku 102-8357, Japan; 4Faculty of Health and Sport Sciences, University of Tsukuba, 1-1-1 Tennodai, Tsukuba 305-8574, Japan

**Keywords:** menstrual-related symptoms, primary dysmenorrhea, premenstrual syndrome, risk factors

## Abstract

This study aimed to identify factors associated with the prevalence and severity of menstrual-related symptoms. The protocol was registered in PROSPERO (CRD42021208432). We conducted literature searches of PubMed and Ichushi-Web and used the Jonna Briggs Institute critical appraisal checklist to assess the quality. Of the 77 studies included in the meta-analysis, significant odds ratios (ORs) were obtained for eight factors associated with primary dysmenorrhea (PD): age ≥ 20 years (OR: 1.18; 95% confidence interval [CI]: 1.04–1.34), body mass index (BMI) < 18.5 kg/m^2^ (OR: 1.51; 95% CI: 1.01–2.26), longer menstrual periods (OR: 0.16; 95% CI: 0.04–0.28), irregular menstrual cycle (OR: 1.28; 95% CI: 1.13–1.45), family history of PD (OR: 3.80; 95% CI: 2.18–6.61), stress (OR: 1.88; 95% CI: 1.30–2.72), sleeping hours < 7 h (OR: 1.19; 95% CI: 1.04–1.35), and bedtime after 23:01 (OR: 1.30; 95% CI: 1.16–1.45). Two factors were associated with severity of PD (moderate vs. severe): BMI < 18.5 kg/m^2^ (OR: 1.89; 95% CI: 1.01–3.54) and smoking (OR: 1.94; 95% CI: 1.08–3.47). PD severity (mild vs. severe) and prevalence of premenstrual syndrome were associated with BMI < 18.5 kg/m^2^ (OR: 1.91; 95% CI: 1.04–3.50) and smoking (OR: 1.86; 95% CI: 1.31–2.66), respectively. The identified risk factors could be utilized to construct an appropriate strategy to improve menstrual symptoms and support women’s health.

## 1. Introduction

Menstruation is a physiological phenomenon that specifically occurs in women. In a healthy woman, it is regulated by cyclical fluctuations of female hormones such as estrogen and progesterone, which occur regularly during a cycle of about one month [1]. During the menstrual cycle, from the first day of menstruation to the day preceding the next menstrual period, various changes occur in a woman’s body caused by major hormonal fluctuations. Therefore, menstrual-related symptoms are one of the most common problems faced by women [1]. In particular, primary dysmenorrhea (PD) and premenstrual syndrome (PMS) are reported to cause symptoms in many women with considerable impact on their daily life [2,3,4]. Therefore, the improvement of menstrual-related symptoms should lead to better quality of life for a vast number of women. Many studies have investigated the factors associated with the prevalence and severity of menstruation-related symptoms [5,6]. Faramarzi et al. [5] reported that women with higher stress levels and higher caffeine intake exhibited a high rate of dysmenorrhea but revealed no association with physical activity or breakfast skipping. Hu et al. [6] reported that women with short sleep duration, late bedtime, and those who skipped breakfast had a high rate of dysmenorrhea, but physical activity and caffeine intake were not associated. Thus, there is no consensus regarding the factors associated with the prevalence and severity of menstrual-related symptoms. Therefore, we aimed to identify factors associated with the prevalence and severity of menstrual-related symptoms by conducting a systematic review and meta-analysis. The significance of this study is that by identifying the factors associated with the prevalence and severity of menstrual-related symptoms, an appropriate strategy to improve menstrual-related symptoms and support women’s health could be developed.

## 2. Materials and Methods

This systematic review was conducted following the Preferred Reporting Items for Systematic Reviews and Meta-analyses (PRISMA) protocols [7]. The study protocol was registered in the PROSPERO database (registration ID: CRD42021208432).

### 2.1. Study Selection

A literature search was conducted using PubMed and Ichushi-Web (an exhaustive collection of Japanese biomedical literature) (Table 1). The search was performed on 15 January 2021. The following inclusion criteria were applied: (1) written in English and Japanese; and (2) observational studies (cross-sectional studies, cohort studies, and case-control studies). The following exclusion criteria were applied: (1) the participants were not human; (2) the participants included men; (3) the participants included women with diseases, pregnant women, and premenopausal or postmenopausal women; (4) the studies did not describe the characteristics of the participants; (5) the studies did not assess PD and PMS; and (6) the studies did not examine factors related to the prevalence and severity of PD and PMS.

Studies were independently screened by two reviewers (R.M. and A.S.), first by title and abstract, and next by full text according to the inclusion and exclusion criteria. In case of discrepancies in the screening results, a third reviewer (Y.N.) was added to the discussion and the issue was resolved by consensus among the three reviewers.

### 2.2. Data Extraction

The two independent reviewers extracted the following data from the eligible studies: name of first author; publication year; country; study design; sample size; and sample characteristics, including age, outcome measures, and exposure outcomes.

### 2.3. Quality Assessment

The two independent reviewers evaluated the quality of the included study according to the Jonna Briggs Institute (JBI) critical appraisal checklist [8]. The JBI evaluates bias by selecting the closest answer to a specific question from Yes, No, unclear, or not applicable. On completion of the evaluations, the JBI score, which comprised the percentage of “yes” responses, was calculated for each study. The studies with a JBI score ≥ 50% and <70% were considered as having middle quality, while the studies with a JBI score ≥ 70% were considered as having high quality; these studies with middle or high quality were included in our meta-analysis. In cases of discrepancies in the results, the third reviewer was added to the discussion and the issue was resolved by consensus among the three reviewers.

### 2.4. Statistical Analysis

Statistical analysis and meta-analysis were performed using Review Manager software version 5.4 (The Nordic Cochrane Centre, Copenhagen, Denmark). We used the random-effects models to calculate each outcome’s risk regarding prevalence and severity (mild, moderate, or severe) for PD and PMS. The effect size was reported as odds ratio (OR) and its 95% confidence interval (CI).

## 3. Results

### 3.1. Study Selection

A flow diagram outlining the study selection in this systematic review and meta-analysis is presented in Figure 1. A total of 1479 studies were retrieved in the database search: 1299 from PubMed, and 180 from Ichushi-Web. Of these studies, three were duplicated.

After screening the titles and abstracts of 1476 studies, 1108 that did not meet the inclusion criteria were excluded. After screening the full text of 368 studies, 237 were excluded. The risk of bias of the remaining 131 studies was evaluated using the JBI score (Appendix A); among these, 31 (JBI score ≥ 70%) [5,9,10,11,12,13,14,15,16,17,18,19,20,21,22,23,24,25,26,27,28,29,30,31,32,33,34,35,36,37,38], 46 (JBI score ≥ 50% and <70%) [6,39,40,41,42,43,44,45,46,47,48,49,50,51,52,53,54,55,56,57,58,59,60,61,62,63,64,65,66,67,68,69,70,71,72,73,74,75,76,77,78,79,80,81,82,83], and 54 (JBI score < 50%) [84,85,86,87,88,89,90,91,92,93,94,95,96,97,98,99,100,101,102,103,104,105,106,107,108,109,110,111,112,113,114,115,116,117,118,119,120,121,122,123,124,125,126,127,128,129,130,131,132,133,134,135,136,137] were of high, middle, and low quality, respectively. Hence, 54 studies were excluded. Finally, 77 studies were included in the meta-analysis.

### 3.2. Characteristics of the Included Studies

Of the 77 studies, 44 were cross-sectional, 30 were case-control, and 3 were cohort studies. Moreover, 31 examined PD, 26 examined PMS, 15 examined PMS and premenstrual dysphoric disorder (PMDD), and 3 examined PMDD. The remaining two studies examined factors of PD and PMS as well as PD, PMS, and PMDD, respectively. 

### 3.3. Primary Dysmenorrhea

#### 3.3.1. Characteristics of Reporting of PD

The timing of the onset of PD symptoms was reported in seven studies [6,13,45,59,64,66,68] (Table 2). The proportion of symptoms occurring from the first day of menstruation was the highest in all seven studies (31.6–70.0%). The timing of the onset of PMS symptoms was not reported.

The duration of PD was reported in eight studies [6,13,23,44,53,59,66,68]. Among them, two studies [23,44] reported the mean ± standard deviation (SD). Three more studies [6,59,68] reported the proportion, and the others reported the most frequent days [13,53,66], the mean days [13,53,66], and the median and 25th/75th percentile [13,53,66] (Table 3).

The effects of PD were reported in five studies [45,66,68,73,83] (Table 4). The effect of PD on work or school was reported in four studies (4.4–44.7%). The effect of PD on daily life was reported in three studies (54.5–92.4%), and of these two studies reported that the impact on daily life was highest.

The methods of coping with symptoms of PD were reported in four studies [21,45,68,75] (Table 5). All four studies reported the use of analgesics (32.5–67.0%), and in three of these studies, the use of analgesics (39.9–67.0%) was highest.

Details of the symptoms of PD were reported in 10 studies [5,6,30,38,44,45,58,68,72,75], and 9 of the 10 studies reported the prevalence [5,6,30,38,44,58,68,72,75] (Appendix A). The following 17 symptoms were reported in descending order of frequency: nausea, fatigue, abdominal pain, diarrhea, headache, low back pain, irritability, dizziness, decreased concentration, changes in sleep, depression, anxiety-tension, mood depression, changes in appetite, inflammation of the skin, edema, and lack of interest.

#### 3.3.2. Risk Factors for PD

From the 77 studies, we extracted data regarding (1) physical characteristics, such as age, body mass index (BMI), and race; (2) menstrual characteristics, such as age at menarche, number of menstrual days, menstrual cycle, and family history of PD; and (3) lifestyle factors, such as stress, smoking habit, alcohol intake, caffeine intake, skipping breakfast, physical activity, sleep duration, and bedtime. The meta-analysis revealed that eight factors were significantly associated with the prevalence of PD: age ≥ 20 years (OR: 1.18; 95% CI: 1.04–1.34) (Figure 2); BMI < 18.5 kg/m^2^ (OR: 1.51; 95% CI: 1.01–2.26) (Figure 3); longer menstrual periods (OR: 0.16; 95% CI: 0.04–0.28) (Figure 4); irregular menstrual cycle (OR: 1.28; 95% CI: 1.13–1.45) (Figure 5); family history of PD (OR: 3.80; 95% CI: 2.18–6.61) (Figure 6); stress (OR: 1.88; 95% CI: 1.30–2.72) (Figure 7); sleeping for <7 h (OR: 1.19; 95% CI: 1.04–1.35) (Figure 8); and bedtime after 23:01 (OR: 1.30; 95% CI: 1.16–1.45) (Figure 9). A significant OR was found for two factors associated with the severity of PD (severe vs. moderate): BMI < 18.5 kg/m^2^ (OR: 1.89; CI: 1.01−3.54) (Figure 10) and smoking habit (OR: 1.94; 95% CI: 1.08–3.47) (Figure 11). The severity of PD (severe vs. mild) was significantly associated with BMI < 18.5 kg/m^2^ (OR: 1.91; 95% CI: 1.04–3.50) (Figure 12).

### 3.4. Premenstrual Syndrome

#### 3.4.1. Characteristics of Reporting of PMS

The timing of the onset of PMS symptoms was not reported. The duration of PMS was reported in one study [50]. This study reported that the mean ± standard deviation (SD) was 7.1 ± 3.5 (days).

The effects of PMS were reported in two studies [54,73] (Table 6). These two studies reported the effect of PMS on work or school (42.9−44.7%). Among these, one study [54] reported the effect of PMS on daily life (71.3%) and relationships (48.0%), in addition to the impact on work and school, the highest impact on daily life.

The methods of coping with symptoms of PMS were reported in one study [67]. This study reported the use of analgesics (45.6%), warming (25.0%), rest (21.9%), and exercise (7.9%). The use of analgesics was highest.

Details of the symptoms of PMS were reported in 18 studies [11,17,18,19,27,30,32,39,40,46,50,54,56,57,62,63,73,90], and 14 of the 18 studies reported the prevalence [11,17,18,19,27,30,39,40,46,50,54,57,63,73] (Appendix A). The following 24 symptoms were reported in the descending order of frequency: irritability, fatigue, anxiety-tension, fatigue, changes in appetite, changes in sleep, breast tenderness, headache, decreased concentration, abdominal pain, low back pain, tearfulness, inflammation of the skin, edema, decreased interest, abdominal distension, nausea, mood elevation, cramps, depressed mood, diarrhea, muscle tension, hot flashes, and dizziness. 

#### 3.4.2. Risk Factors for PMS

From the 77 studies, we extracted data regarding (1) physical characteristics, such as age, body mass index (BMI), and race; (2) menstrual characteristics, such as age at menarche, number of menstrual days, menstrual cycle, and family history of PD; and (3) lifestyle factors, such as stress, smoking habit, alcohol intake, caffeine intake, skipping breakfast, physical activity, sleep duration, and bedtime. The meta-analysis revealed that the prevalence of PMS was significantly associated with smoking (OR: 1.86; 95% CI: 1.31–2.66) (Figure 13).

## 4. Discussion

### 4.1. Principal Findings

This study aimed to identify factors associated with the prevalence and severity of menstrual-related symptoms. As a result, significant ORs were observed for some of the physical characteristics, menstrual characteristics, and lifestyle factors.

Regarding physical characteristics, age and BMI were significantly associated with the prevalence of PD, and BMI was associated with the severity of PD. PD is known to be more common during the late teens compared to during the early twenties, peaking around age 20 and decreasing thereafter [138]. Two of the studies included in the present meta-analysis, both of which involved university students in their teens and twenties, revealed that the prevalence of PD decreased with increasing age. This supports the findings of a previous study [138]. Considering low BMI values, it is known that normal reproductive development and menarche require adequate body fat mass. A previous study reported that a body fat percentage of at least 17% should be maintained for the onset of menarche and at least 22% is required for regular menstruation [139]. In addition, low available energy, which is associated with low BMI, leads to the disruption of the hypothalamus–pituitary system [139,140]. Disorders of the hypothalamus–pituitary system cause amenorrhea and abnormal menstruation by inhibiting the secretion of luteinizing hormone and follicle-stimulating hormone from the pituitary gland and blocking stimulation of the ovaries [139,140]. Furthermore, Ju et al. [141] reported a U-shaped relationship between the body fat percentage and the prevalence of PD; high BMI has been reported to cause early menarche, irregular menstrual cycles, rare menstruation, amenorrhea, and chronic anovulation [142]. This suggests that, in addition to low BMI, high BMI may also increase the prevalence and severity of PD. Therefore, maintaining an appropriate BMI may reduce the prevalence and severity of PD.

Regarding the characteristics of menstruation, the number of menstrual days, menstrual cycle, and family history of PD were significantly associated with the prevalence of PD. PD is known to result from uterine contractions caused by pain-inducing prostaglandins [143]. Prostaglandins are normally secreted during menstruation to expel the endometrium. However, prostaglandins also cause local vasoconstriction, and excessive prostaglandin secretion causes uterine ischemia and pain, resulting in PD [143]. In addition, longer menstrual periods increase the duration of prostaglandin secretion, which may lead to longer durations and more severe pain. The disruption of the hypothalamus–pituitary system and the blockage of ovarian stimulation cause discontinuation of estrogen production. When the production of estrogen is disrupted, the balance of female hormones is disturbed, resulting in an irregular menstrual cycle [139,144]. Pertaining to family history of PD, Jahanfar et al. [105] examined the genetic factors in menstruation-related problems in monozygotic and dizygotic twins and reported that 40−50% were related to genetic factors. It may also be relevant that approximately 56% of information sources regarding girls’ pain management and menstruation are mothers [145].

For lifestyle factors, stress, sleep duration, and sleeping time were significantly associated with the prevalence of PD. Smoking was also associated with the prevalence of PMS. Female reproductive organs are believed to be highly sensitive to stress, and intense stress could have detrimental effects on health [40,146]. Activation of the corticotropin-releasing hormone system by elevated stress levels inhibits the release of follicle-stimulating hormone and luteinizing hormone, thereby inhibiting follicle development [40,146]. Stress-related hormones such as adrenaline and cortisol can also increase prostaglandin synthesis and cause PD [146]. Additionally, short sleep duration was suggested to lead to a decrease in melatonin secretion [147]. Regarding the effects of melatonin on sexual function, a pituitary−gonadal function is reduced during winter, when melatonin secretion is lower [148], and melatonin administration to neonates causes sexual prematurity [149,150]. Therefore, melatonin secretion may affect reproductive function. In contrast, the occurrence of PD itself was reported to reduce sleep duration; therefore, a causal relationship is unclear [20].

The present study revealed no significant ORs for physical activity, alcohol consumption, caffeine intake, and breakfast skipping. 

Physical activity promotes the release of endorphins in the blood that reduces pain, exerts a relaxing effect on muscle [151], reduces cortisol levels, and reduces the synthesis of prostaglandins that cause pain [152]. For these reasons, physical activity has been shown to be effective in improving PD, suggesting an association between physical activity and PD [151]. However, no association was identified in the present study. The reason could be attributed to differences in exercise intensity. The results of the present study by age group indicated that more studies involving teenagers and participants in the twenties suggested an association with PD and PMS than studies targeting those aged ≥30 years. This may be attributed to the fact that the definition of exercise differed according to the age group. Especially, studies of women in their teens and twenties defined high-intensity exercise, such as performing sports club activities, while studies of women in their thirties and older defined and included low-intensity exercises, such as walking. In a previous study that examined factors associated with the prevalence of PD in female athletes and non-athletes, only non-athletes showed an association between exercise and the prevalence of PD, and the reason for this association might have been exercise intensity [153]. Therefore, exercise intensity should be considered when examining the association between physical activity or exercise and prevalence and severity of PD and PMS. 

For alcohol and caffeine intake, over-contraction of the uterus has been shown to decrease blood flow in the uterus and cause pain [154,155]. Caffeine has been demonstrated to increase uterine contractions [156], and alcohol consumption reduces uterine arterial blood flow by approximately 40% [157]. However, no significant ORs were identified in the meta-analysis of the present study. This difference could be attributed to varying definitions of alcohol and caffeine consumption. Specifically, among the studies included in the present meta-analysis, the criterion of a study [6] of Chinese university students was one glass of wine per month in the past 6 months (approximately 250 mL), whereas the criterion for having alcohol consumption in a study including Turkish university student was ≥1 drink/week [56]. 

Regarding caffeine intake, one criterion was ≥300 mg of caffeinated soft drinks, coffee, decaf coffee, black tea, chocolate milk, and chocolate bars in a study involving Iranian university students [5], whereas in another study involving Chinese university students, the criterion was the consumption of three or more caffeine-containing substances, such as coffee, tea, and chocolate in the past month [6]. These differences in definitions may have led to the conflicting results.

Skipping breakfast also leads to a lower intake of certain foods, and it is suggested that nutritional imbalances may lead to PD [20]. In addition, dairy products and dietary fiber, which are commonly consumed at breakfast, show a significant association with PD, and a lower intake of these nutrients may increase the risk of PD [84,158,159]. Additionally, it was noted that missing breakfast may interfere with diurnal rhythms and negatively affect reproductive function [159], and there may be a link between eating habits and the development of PD and PMS. However, no significant ORs were identified in the present meta-analysis. The first reason for this discrepancy could be the effect of cultural background. Three studies were included in the meta-analysis regarding skipping breakfast: one study of Iranian university students [5], another of Chinese university students [6], and the other of Japanese junior high school students [20]. These cultural differences may be related to the lack of association. Second, each study had a different definition of skipping breakfast. The study of Iranian university students defined skipping breakfast as having breakfast ≤ 1 time per week, whereas the definition in the study involving Chinese university students was not eating breakfast more than once in the past week; the definition used in the study of Japanese junior high school students was occasional or no breakfast intake. Therefore, in the future, consideration of the cultural background with the associated definition of skipping breakfast is necessary to assess the association of each item with the occurrence and severity of symptoms associated with menstruation.

### 4.2. Strengths and Limitations

Many studies have investigated the factors associated with menstrual symptoms in the past few decades. Most of them focus on dysmenorrhea occurring during menstruation [5,6,10,20,28,34,38,58,66,68,83] or premenstrual syndrome occurring before menstruation [32,43,70]. However, many women experience symptoms associated with menstruation both before and during menstruation, rather than only before or during menstruation. Moreover, the discrimination between premenstrual and menstrual symptoms was suggested to be meaningless because the types of symptoms were similar [160]. Furthermore, several factors were associated with the development of menstrual symptoms, and the degree of influence of these factors should be clarified. Therefore, the present systematic review and meta-analysis tried to synthesize the findings of previous publications on pre-menstrual syndromes, which occurs before menstruation, and dysmenorrhea, which occurs during menstruation. To our knowledge, our study is the first to identify factors associated with the prevalence and severity of these symptoms. 

Nevertheless, this study has several limitations. First, many of the symptoms associated with menstruation are self-reported and may include conditions such as organic dysmenorrhea. Of the 34 studies [5,6,9,10,12,20,22,23,24,28,29,30,32,34,35,37,38,41,43,49,55,58,59,60,61,65,66,68,69,70,72,75,80,83] from which meta-analysis data could be extracted for this study, 21 studies [5,6,9,12,20,29,30,32,34,37,38,43,58,59,66,68,70,72,75,80,83] used a self-response questionnaire, and PD and PMS were assessed by the participant, without being examined by a physician. Second, causality of the factors associated with menstrual symptoms is unclear. Many of the studies accepted for inclusion in the systematic review of this study were cross-sectional studies and did not examine causality. For case-control studies [10,22,23,24,28,33,35,41,49,55,60,61,65], only cross-sectional data at the start of the study were extracted. Third, racial and ethnic differences are not considered. The study participants of the included studies in the present systematic review were predominantly from the United States and Turkey. Finally, only 34 of the 77 studies were included in the meta-analysis because we could extract comparable data for the meta-analysis concerning three topics: physical characteristics, menstrual characteristics, and lifestyle. Each analysis had a small number of studies. Therefore, generalizability of the present study is limited.

### 4.3. Future Direction

The present study enabled the identification of factors associated with the prevalence and severity of menstrual-related symptoms. Among these, BMI, stress, sleep duration, and bedtime, which were found to be associated with the prevalence of PD; BMI, which was found to be associated with the severity of PD; and smoking, which was found to be associated with PMS, are items that can be improved through one’s own behaviors. Therefore, the results of this study may help women review their own lives according to their individual symptoms and choose behaviors to improve these symptoms.

## 5. Conclusions

The present systematic review and meta-analysis were performed to identify factors associated with the prevalence and severity of PD and PMS. The results suggest that physical characteristics, such as age and BMI; menstrual characteristics, such as longer menstrual periods, irregular menstrual cycle, and family history of PD; and lifestyle factors, such as smoking, sleeping hours, bedtime, and stress were associated with the prevalence and severity of menstrual-related symptoms.

## Figures and Tables

**Figure 1 ijerph-20-00569-f001:**
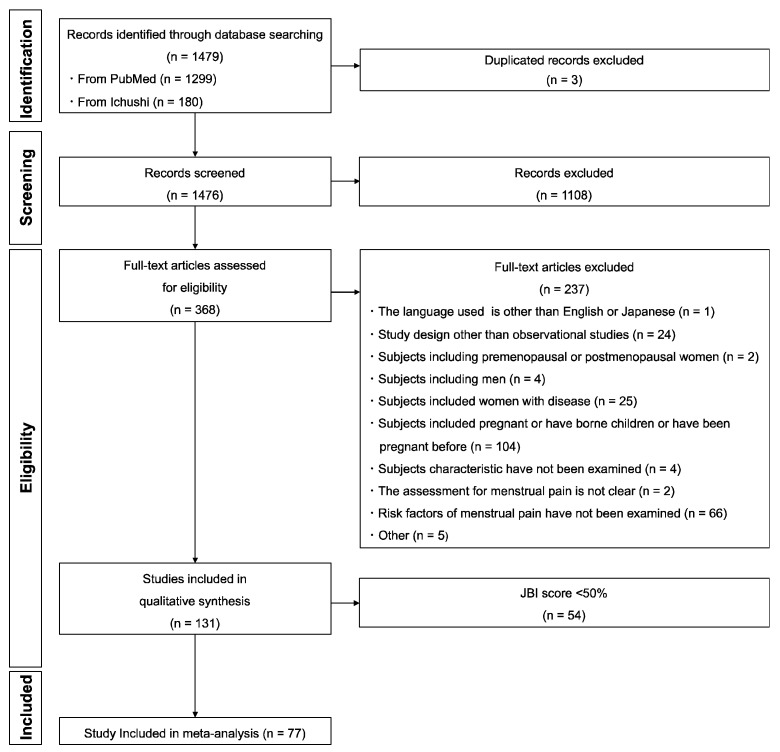
Flow diagram of literature search.

**Figure 2 ijerph-20-00569-f002:**
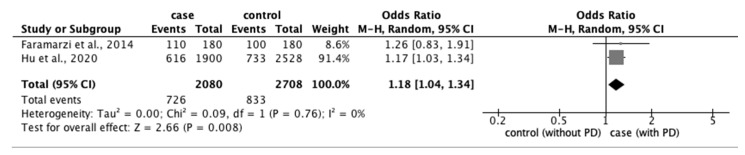
Forest plot showing a meta-analysis of studies comparing the number of women aged ≥20 years with and without PD [5,6]. The analysis was performed using a random-effects model. PD, primary dysmenorrhea. The black diamond and its extremities indicate the pooled risk ratio center and a 95% confidential interval.

**Figure 3 ijerph-20-00569-f003:**
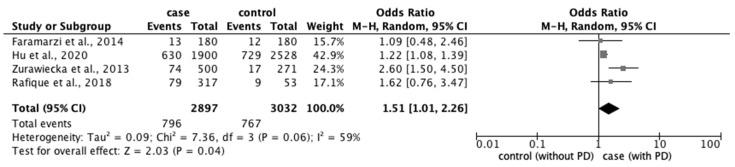
Forest plot showing a meta-analysis of studies comparing the number of women with BMI < 18.5 kg/m^2^ with and without PD [5,6,38,68]. The analysis was performed using a random-effects model. PD, primary dysmenorrhea. The black diamond and its extremities indicate the pooled risk ratio center and a 95% confidential interval.

**Figure 4 ijerph-20-00569-f004:**
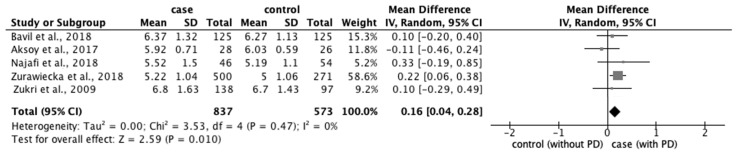
Forest plot showing a meta-analysis of studies comparing menstrual days with and without PD [9,10,28,38,83]. The analysis was performed using a random-effects model. PD, primary dysmenorrhea. The black diamond and its extremities indicate the pooled risk ratio center and a 95% confidential interval.

**Figure 5 ijerph-20-00569-f005:**
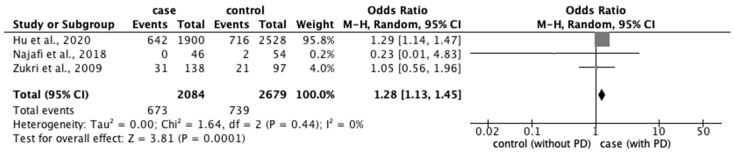
Forest plot showing a meta-analysis of studies comparing the number of women who have irregular menstrual cycles with and without PD [6,28,83]. The analysis was performed using a random-effects model. PD, primary dysmenorrhea. The black diamond and its extremities indicate the pooled risk ratio center and a 95% confidential interval.

**Figure 6 ijerph-20-00569-f006:**
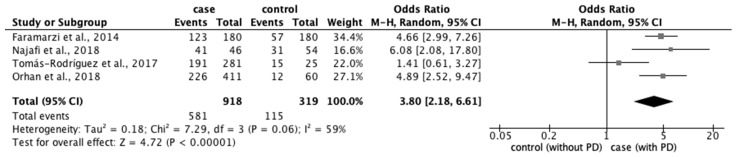
Forest plot showing a meta-analysis of studies comparing the number of women who have a family history of PD with and without PD [5,28,34,66]. The analysis was performed using a random-effects model. PD, primary dysmenorrhea. The black diamond and its extremities indicate the pooled risk ratio center and a 95% confidential interval.

**Figure 7 ijerph-20-00569-f007:**
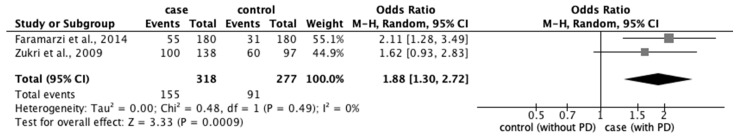
Forest plot showing a meta-analysis of studies comparing the number of women with and without PD who have higher stress levels [5,83]. The analysis was performed using a random-effects model. PD, primary dysmenorrhea. The black diamond and its extremities indicate the pooled risk ratio center and a 95% confidential interval.

**Figure 8 ijerph-20-00569-f008:**
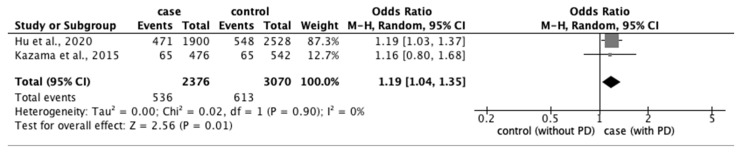
Forest plot showing a meta-analysis of studies comparing the number of women with and without PD sleeping < 7 h [6,20]. The analysis was performed using a random-effects model. PD, primary dysmenorrhea. The black diamond and its extremities indicate the pooled risk ratio center and a 95% confidential interval.

**Figure 9 ijerph-20-00569-f009:**
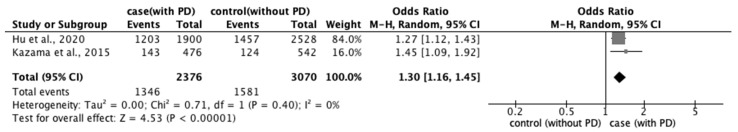
Forest plot showing meta-analysis of studies comparing the number of women with and without PD who go to bed at 23:01 or later [6,20]. The analysis was performed using a random-effects model. PD, primary dysmenorrhea. The black diamond and its extremities indicate the pooled risk ratio center and a 95% confidential interval.

**Figure 10 ijerph-20-00569-f010:**
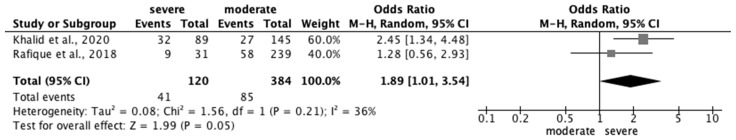
Forest plot showing a meta-analysis of studies comparing the number of women with BMI < 18.5 kg/m^2^ by the severity of PD (severe vs. moderate) [58,68]. The analysis was performed using a random-effects model. PD, primary dysmenorrhea. The black diamond and its extremities indicate the pooled risk ratio center and a 95% confidential interval.

**Figure 11 ijerph-20-00569-f011:**
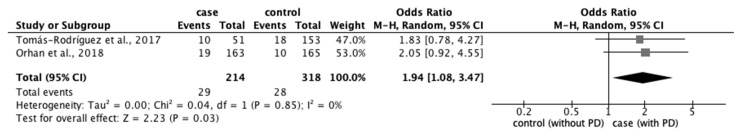
Forest plot showing a meta-analysis of studies comparing the number of women with a smoking habit by the severity of PD (severe vs moderate) [34,66]. The analysis was performed using a random-effects model. PD, primary dysmenorrhea. The black diamond and its extremities indicate the pooled risk ratio center and a 95% confidential interval.

**Figure 12 ijerph-20-00569-f012:**
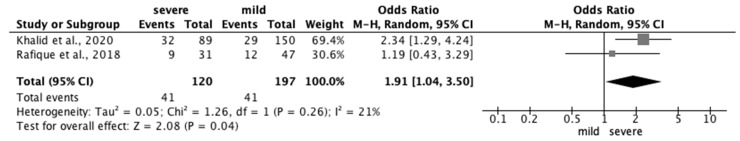
Forest plot showing a meta-analysis of studies comparing the number of women with BMI < 18.5 kg/m^2^ by the severity of PD (severe vs. mild) [58,68]. The analysis was performed using a random-effects model. PD, primary dysmenorrhea. The black diamond and its extremities indicate the pooled risk ratio center and a 95% confidential interval.

**Figure 13 ijerph-20-00569-f013:**
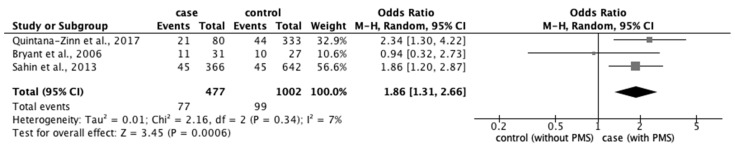
Forest plot showing a meta-analysis of studies comparing the number of women with and without PMS with a smoking habit [32,43,70]. The analysis was performed using a random-effects model. PMS, premenstrual syndrome. The black diamond and its extremities indicate the pooled risk ratio center and a 95% confidential interval.

**Table 1 ijerph-20-00569-t001:** Database search strategies.

Databases	Search Strategies (January 2021)
PubMed	((“premenstrual syndrome” [All Fields]) OR (“premenstrual dysphoric disorder” [All Fields]) OR (“functional dysmenorrhea” [All Fields]) OR (“primary dysmenorrhea” [All Fields]) OR (“menstrual pain” [All Fields]) OR (“menstrual symptoms” [All Fields])) AND ((“risk factors” [All Fields]) OR (“severity” [All Fields]) OR (“exacerbation” [All Fields]) OR (“limitation” [All Fields]) OR (“activity of daily life” [All Fields]) OR (“limiting factors” [All Fields]))
Ichushi	(“premenstrual syndrome” OR “premenstrual dysphoric disorder” OR “functional dysmenorrhea” OR “primary dysmenorrhea” OR “menstrual pain” OR “menstrual symptoms”) AND (“risk factors” OR “severity” OR “exacerbation” OR “limitation” OR “activity of daily life” OR “limiting factors”)

**Table 2 ijerph-20-00569-t002:** Onset of primary dysmenorrhea (%).

First Author (Year)	Before Menstruation	First Day of Menstruation	on or after the Second Day of Menstruation
Hu et al. (2020) [6]	24.5	60.8	14.7
Balbi et al. (2000) [13]	70.0		N/A
Chen et al. (2005) [45]	N/A	49.8	N/A
Kural et al. (2015) [59]	23.5	61.5	14.2
Monday et al. (2019) [64]	N/A	31.6	N/A
Orhan et al. (2018) [66]	35.0	65.0	N/A
Rafique et al. (2018) [68]	44.5	40.1	15.5

N/A, not applicable.

**Table 3 ijerph-20-00569-t003:** Duration of primary dysmenorrhea and premenstrual syndrome.

First Author (Year)	Mean ± SD	1 Day	2 Days	3 Days	4 Days	Over 5 Days
Hu et al. (2020) [6]	N/A	N/A	95.3	4.7	N/A	N/A
Balbi et al. (2000) [13]	N/A	N/A	Most frequent	N/A	N/A	N/A
Liu et al. (2017) [23]	1.9 ± 0.38 (days)	N/A	N/A	N/A	N/A	N/A
Carman et al. (2018) [44]	4.37 ± 5.55 ^a^ (hours)	N/A	N/A	N/A	N/A	N/A
Harlow et al. (1996) [53]	1.7 ^b^ (days)	N/A	N/A	N/A	N/A	N/A
Kural et al. (2015) [59]	N/A	37.0	39.8	15.4	5.4	2.3
Orhan et al. (2018) [66]	2.0 [1.0–3.0] ^c^ (days)	N/A	N/A	N/A	N/A	N/A
Rafique et al. (2018) [68]	N/A	25.2	55.5	N/A	N/A	N/A

N/A, not applicable. ^a^ Data are presented as mean durations of menstruation-related headache. ^b^ SD was not shown. ^c^ Data are presented as medians [25th/75th percentile].

**Table 4 ijerph-20-00569-t004:** Effects of primary dysmenorrhea (%).

First Author (Year)	Daily Life	Work or School	Sports Activities	Social Activities	Less Concentration	Felt More Rested	Avoidance of Responsibility
Chen et al. (2005) [45]	92.4	25.3	N/A	N/A	N/A	N/A	N/A
Orhan et al. (2018) [66]	N/A	4.4	4.5	4.5	N/A	N/A	N/A
Rafique et al. (2018) [68]	54.5	N/A	25.2	N/A	N/A	N/A	N/A
Tangchai et al. (2004) [75]	N/A	21.1	37.3	18.2	63.6	N/A	N/A
Zukri et al. (2009) [83]	88.2	31.1	N/A	81.2	N/A	99.9	92.1

N/A, not applicable.

**Table 5 ijerph-20-00569-t005:** Methods of coping with primary dysmenorrhea (%).

First Author (year)	Analgesics	Warming	Rest	Exercise	Medical Treatment	Herbal Medicine	Meditating
Kordi et al. (2013) [21]	67.0	N/A	N/A	N/A	N/A	N/A	N/A
Chen et al. (2005) [45]	39.9	N/A	N/A	N/A	N/A	N/A	N/A
Rafique et al. (2018) [68]	63.7	18.3	N/A	N/A	N/A	N/A	N/A
Tangchai et al. (2004) [75]	32.5	34.0	92.0	6.8	7.1	12.7	4.5

N/A, not applicable.

**Table 6 ijerph-20-00569-t006:** Effects of premenstrual syndrome (%).

First Author (Year)	Daily Life	Work or School	Relationships
Hashim et al. (2019) [54]	71.3	44.7	48.0
Tadakawa et al. (2016) [73]	N/A	42.9	N/A

N/A, not applicable.

## Data Availability

Not applicable.

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
