# Peer review of "Factors Associated with the Prevalence and Severity of Menstrual-Related Symptoms: A Systematic Review and Meta-Analysis"

_ijerph, 2022, doi:10.3390/ijerph20010569_

Round 1

Reviewer 1 Report

The manuscript Factors associated with the prevalence and severity of menstrual-related symptoms: a systematic review and meta-analysis?” aimed to identify factors associated with the prevalence and severity of menstrual related symptoms e.g., primary dysmenorrhea and premenstrual symptoms in a systematic review and meta-analysis. This has not previously been performed and will contribute to new knowledge. However, as the authors state as a limitation, most of the included studies have used self-reported data and most studies were predominantly performed in the United States and Turkey, which reduces the generalizability. Even so, the article fills a gap in the literature.

Over all comments:

The introduction gives the reader a good overview of the topic.

The methods are well described.

The discussion section is well written with strengths and limitations of the study.

Specific comments:

-It would be easier to the reader if the results were divided in sections about primary dysmenorrhea and premenstrual symptoms. It is a bit confusing to read the tables as it is presented now.

-Table 2: Is the proportion in the reference Babi et al. correct? 70 % for both Before menstruation and First day of menstruation?

-In the tables when there is a –  Is it the same as non-applicable?

-In the meta-analysis only 34 of the 77 articles were included and you extracted data regarding three different topics: physical characteristics, menstrual characteristics, and lifestyle factors. This results in few studies in each topic. It would be valuable to include this in the limitation section and discuss how this affected the results.

The topic of the review and meta-analysis is important and after some clarifications and revisions, it would be a valuable contribution to the literature.

Reviewer 2 Report

Dear author's

I was pleased to review your manuscript. The subject is not new but the article is very well written.

Knowing this results there are some clinical implication that you want to propose?

Please explain the novelty of this study.
